# A Study on the Algorithm of Quality Evaluation for Fiber Laser Welding Process of ASTM A553-1 (9% Nickel Steel) Using Determination of Solidification Crack Susceptibility

**DOI:** 10.3390/ma13245617

**Published:** 2020-12-09

**Authors:** Minho Park, Jisun Kim, Changmin Pyo, JoonSik Son, Jaewoong Kim

**Affiliations:** 1Southwestern Branch Institute, Research Institute of Medium & Small Shipbuilding, Jeonnam 58457, Korea; mhpark@rims.re.kr (M.P.); jsson@rims.re.kr (J.S.); 2Smart Manufacturing Process R&D Group, Korea Institute of Industrial Technology, Gwangju 61012, Korea; kimjisun@kitech.re.kr (J.K.); changmin@kitech.re.kr (C.P.)

**Keywords:** fiber laser welding, ASTM A553-1 (9% nickel steel), discriminant analysis, solidification crack susceptibility, optimization

## Abstract

To prevent the contamination of the marine environment caused by ship exhaust gas, the demand for LNG (liquefied natural gas) fueled ships is increasing worldwide. A tank to store LNG at cryogenic temperatures is indispensable to such LNG-fueled ships. Since the materials used for LNG fuel propulsion tanks must have excellent mechanical properties such as impact toughness at cryogenic temperatures, the International Maritime Organization limits the IGC Code only to four types. Most of the tank materials for LNG-fueled ships ordered recently are adopting ASTM A553-1 material, but a systematic study to analyze the problem of quality degradation that may occur when welding A553-1 steel is required to secure the safety of cryogenic tanks. Therefore, in this study, among various quality problems, the tendency of weld solidification crack vulnerability is identified, and a decision system and optimization procedure are developed. In addition, a method of securing the welding quality of A553-1 steel was proposed by setting quality deterioration standards.

## 1. Introduction

The International Maritime Organization (IMO) enforces strict regulations on the emissions of ships and it is expected that the regulations will continue to be strengthened, for example by strictly controlling the emission of pollutants in the sea. These marine environmental regulations are causing new changes in the shipping and shipbuilding industries. To cope with the regulations by IMO, LNG (liquefied natural gas) fueled ships (LFS) were developed, using LNG as a fuel, and are being operated in Northern Europe [1,2,3,4]. ASTM A553-1 (9% nickel steel) is usually used as the steel to manufacture LNG storage tanks and the A553-1 steel has a high tensile strength and is usually used for a tank storing cryogenic fuel with QT (quenching tempering) treatment. Because it has excellent impact toughness at cryogenic temperatures and is economically advantageous, there are a number of references as a material for an LNG storage tank. Its impact toughness varies depending on its specifications, but most have an absorbed energy of 34 J or more at −196 °C. In Korea, 9% Ni steel, which has excellent quality characteristics equal to or better than those of developed countries, was developed using the QLT (quenching lamellarizing tempering) method in the 1990s and is being supplied to the LNG storage tank industry in Korea [5,6].

Regarding A553-1 steel welding, welding is difficult because the welding material has a lower melting point than a base material and the welding quality varies depending on the welder’s skills. Therefore, it is necessary to review the problems that may occur during welding of A553-1 steel and to avoid welding defects by evaluating the characteristics of a welding part in a study. When looking at the prior studies related to the quality deterioration of A553-1 steel, Kim [7] studied the laboratory analysis and meta-analysis for the weldability of A553-1 steel used for an LNG storage tank carrier. In the study, he analyzed the prior studies that suggested the effectiveness of A553-1 steel by applying laboratory research and metaresearch, diagnosed the problems of prior studies, and suggested future research directions. In addition, Lee [8] explains the problem of the welding part caused by the magnetization of 9% nickel steel, and predicts that the utilization of 9% nickel steel will increase through automation and overcoming the problem of the future welding process. Kim [9] et al. reported that, when 9% nickel steel was welded by GMAW (gas metal arc welding), the high-temperature cracking susceptibility was good, but when the amount of heat input was increased, the strength of the welded portion decreased. When looking at the current status of research on the design and mechanical properties of cryogenic steel and A553-1 steel used to manufacture LNG storage tanks, Saitoh et al. [10] developed a high Ni-based 9% Ni steel with excellent low-temperature toughness to ensure safety with LNG storage tanks. Watanabe [11] conducted a double tension test on a surface notch of A553-1 steel and reported the reduction of impurities which led to the improvement in fracture toughness and scattering suppression of the toughness characteristics of A553-1 steel. In recent years, optimization through experimental methods has been observed, but research using neural networks and machine learning methods is also becoming active. X Liu et al. [12] conducted a study to measure and analyze the fracture toughness of metals using machine learning (ML) models such as regression trees and neural networks (NN).

Most of the prior studies identified the correlation between welding process parameters and mechanical properties (hardness, tension, impact, etc.) for cryogenic steels such as 9% Ni steel or STS 316L material, etc. and reviewed the problems that may occur when they were applied to LNG storage tanks [13,14]. However, securing the integrity of a welded area of cryogenic steels in the prior studies did not consider the complex factors that can deteriorate the quality of a welded area. Of course, research on process improvement to improve the productivity of cryogenic steel welding has a very important academic meaning, but it remains unknown whether it is possible to secure welding integrity by reflecting the unique process environment of each site, thus resulting in low versatility [15,16].

A typical welding defect that can occur regardless of the process in the melt welding of A553-1 steel is solidification crack. As a columnar crystal grows in a weld metal, a concentration of solute elements in a residual melt occurs and the shrinkage stress at the end of solidification decreases due to the impurities formed between the columnar crystals. Subsequently, this leads to deteriorated strength at the welding part. To improve the quality deterioration of the welding part, it is necessary to optimize the process to avoid the defect by identifying welding process parameters and the characteristics of welding parts. Therefore, this study analyzed the characteristics of quality deterioration of the solidification crack that occurred in a fiber laser welding process applied to 9% Ni steel, which is a cryogenic steel usually used in the LNG storage tank industry, and suggested a proposal to optimize the process parameters to avoid strength decrease in a welding part resulting from the solidification crack.

## 2. Materials and Methods

This welding experiment was performed to check the quality of the laser welding part of A553-1 steel and to identify optimal process parameters. For the experiment, a 5 kW class fiber laser welding machine (Miyachi ML-6950A model, Amada Weld Tech Co. Ltd., Chiba, Japan) was used and the entire system was configured as shown in Figure 1 using a Motoman (Yaskawa DX100 model, Yaskawa Electric Co., Kitakyushu, Japan).

The specimen was wiped clean with ethyl alcohol and sandpaper so that contaminants such as rust, scale, or oxide on the surface of the specimen to be welded did not cause a welding defect. Figure 2 shows a schematic diagram of the fiber laser welding process. The mechanical properties and chemical composition of A553-1 steel used in the welding experiment are shown in Table 1; Table 2, respectively.

Since the fiber laser welding process used in this experiment performs welding while generating keyholes by transmitting high energy required for welding to the material surface, laser power, defocusing, and welding speed were selected as input parameters and the weldability was analyzed by collecting output parameters such as penetration shape and hardness, etc. [11]. Figure 3 shows a schematic diagram to measure the penetration shape of a welding part [17].

In this experiment, the appropriate level or range of input parameters (laser power, defocusing, and welding speed) were selected through a preliminary experiment. A total of 18 experimental conditions were created because there are three levels of laser output, three levels of defocusing, and two levels of welding speed. Table 3; Table 4 show the experimental parameters and levels of input parameters and the total of 18 experimental conditions.

## 3. Results

### 3.1. Measurement of Penetration Geometry

The BOP (bead on plate) fiber laser welding of 9% Ni steel produced proper penetration for the welding process parameters. Moreover, the overall penetration was good and there was no pore or defect in its appearance. To examine the cross-sectional shape of the test specimen, 90% Ethanol + 10% Nitric acid solution was mixed and used to etch its cross section. Then, an optical microscope system was used to accurately measure the penetration shape. Table 5 shows the cross section of welding parts and the measurement results of the penetration area obtained using a 10× optical microscope.

### 3.2. Measurement of Weldment Hardness

A hardness test was performed on each test specimen to check the strength decrease of a welding part that occurs when the impurities with low melting points floated upward due to density difference while the fiber laser welding part was solidified. The Vickers hardness test was performed on the upper part where the impurities appeared, the load used for the hardness test was selected as 0.5 N, and the analysis was performed at the interval of 0.83 mm that did not affect relative hardness. Figure 4 shows a schematic diagram of the hardness test for the welded part of 9% Ni steel. Table 6 and Figure 5 shows the experiment results for the hardness of the upper part of the welded part and heat-affected part, and the hardness test result indicates the average measurement value of five points. The hardness of the upper part shows a value between 264.2 Hv ~ 312.4 Hv, and this value is higher than 243 Hv which is the standard for 9% Ni steel. Therefore, it is judged that sufficient weldability was achieved.

### 3.3. Measurement of Chemical Composition for Weldment

The impurities of Ti, Nb, Mo, and Si that affect crack susceptibility at the penetration surface of the fiber laser welding part were measured and the impurity grain generation tendency that changes according to the welding process parameters was checked. The EDS (energy dispersive spectroscopy) measurement was performed for nine sections as shown in Figure 6. Figure 7 shows the measurement of grains assumed as impurities on the penetration surface and the average value results of the analysis of all four components, such as Ti, Nb, Mo, and Si, are shown in Table 7.

## 4. Discriminant of Quality Characteristics of 9% Ni Steel

### 4.1. Solidification Crack Susceptibility

A Ni-based alloy is composed of residual austenite and lamellar ferrate structure and is basically vulnerable to solidification crack. Therefore, it is very important to solve the problem of solidification cracks when developing the welding process for 9% Ni steel. In addition, the welding process parameters are known to have a significant effect on crack resistance related to solidification cracks. Furthermore, a crack may occur more easily as a welding current and bead/welding rod ratio are higher.

Therefore, the welding current and welding speed should not be too high to obtain a robust welding metal without cracks. In particular, detail management is required for the areas with large constraints such as ultralayer welding. As the importance of solidification crack becomes obvious, Jung-Jin et al., reviewed the susceptibility of solidification crack for a Ni-based alloy generated in melt welding and defined the relationship between impurity elements and crack susceptibility as the solidification crack susceptibility index (*P_SC_*) as shown in Equation (1) [18].
*P*_SC_ = 69.2Ti + 27.3Nb + 9.7Mo + 300Si − 55.3(1)

Therefore, since the fiber laser welding used in this study belongs to a melt welding, the solidification crack susceptibility of 9% Ni steel is reviewed by using the solidification crack susceptibility index proposed above. To evaluate solidification crack susceptibility, we calculated it by using the component analysis values shown in Table 7 and Equation (1).

The purpose of this study was to check the phenomenon where the hardness of the upper welding part decreases due to grain boundary relaxation when crack susceptibility increases due to the increased amount of impurities, and to define the criteria of crack susceptibility. The crack susceptibility of a fiber laser welding process was found to be between 117.0 and 198.6 and it was confirmed that the hardness of the upper welding part was stabilized when the crack susceptibility of a fiber laser welding process was calculated to be 130 or less, as shown in Figure 8.

This stabilization of the upper hardness is because if the data trend falls below 132.2, a drift that can predict the weld hardness is found, and if it is less than 140, the distribution of the upper hardness is randomly distributed and no association is identified.

If impurity elements are distributed with a solidification crack susceptibility of 132.2 or higher, it is judged to be very difficult to secure high-quality hardness of a welding part due to a decrease in density and it can be used as data to define the crack susceptibility criteria for a process.

The solidification crack susceptibility of 132.2 identified above is a standardized score that can be used as an evaluation index for a specific process and indicates that the solidification crack susceptibility of an upper weld starts to appear if it is calculated as a higher score. Based on this, the criteria for solidification crack susceptibility were defined as shown in Table 8. These standardized scores can be used as learning data to determine the decrease in grain boundary strength due to crack susceptibility in the future. This is also to prevent the problem of microcracks caused by impurity grains in the 9% Ni steel welding part to which the corresponding welding process was applied.

### 4.2. Discriminant Analysis

The system to determine the solidification crack susceptibility in a fiber laser welding process of 9% Ni steel is a technique that determines the affiliation of input data by making a model using collected data and entering it into developed group learning data [19,20,21].

The solidification crack susceptibility discriminant system in this study developed a discriminant model using an SVM (Support Vector Machine) technique. SVM is one of the machine learning classification algorithms. Unlike empirical risk minimization based on statistical learning theory, SVM takes a method of reducing errors by using structural risk minimization. In the case of neural networks, as the number of input variables increases, the computation time increases and the separation rate decreases, but in the case of SVM, if mapping to a specific space through a tunnel function, the calculation is simple and classification performance is improved even if the original data has a high number of dimensions. 

Due to these advantages, it is considered to be a very suitable method for this study with many variables, and it was adopted as a method to classify the coagulation crack susceptibility group. The system attempted to determine the possibility of solidification crack susceptibility in the process of addressing the problem of identifying a hyper plane that maximizes a margin within two classes that can be linearly separated. This attempt was based on Equation (2) in the VC (Vapnik–Chervonenkis) theory [22].
*w* · *x* + *b* = 0(2)

*w* is the weight vector, *x* is the input vector, and *b* is the reference value, and the SVM technique described above sequentially performs minimization of complex calculations in the QP (quadratic programming) process. The parameters used for learning in the solidification crack susceptibility discriminant model are welding process parameters (laser power, defocusing, welding speed), penetration shape (penetration width, penetration depth), upper hardness, HAZ (heat affected zone) hardness, and solidification crack susceptibility (*P_SC_*). Therefore, 162 data points were entered with a total of nine multiple parameters. Regarding groups that determine solidification crack susceptibility, the unstable group is defined as 1, and the stable group is defined as 0 to evaluate the predicted discriminant performance according to the SVM technique.

Table 9 shows the learning data to determine a solidification crack susceptibility and Table 10 and Figure 9 quantitatively show a group’s discriminant performance predicted by the data learned in the SVM technique.

## 5. Optimization of Fiber Laser Welding of 9% Ni Steel

### 5.1. Development of Mathematical Model Welding Factors

The response surface analysis method is a statistical analysis method for a response surface where there is a change in response when several input parameters $x_{1},\;x_{2},x_{3},\;\cdots x_{k}$ have a complex action and affect the output parameter y. It is possible to predict how the value of an output parameter changes according to the change in the input parameter value by estimating the functional relationship between input parameters and output parameters from the data and determining which value of an input parameter causes an optimal response amount.

Therefore, this method has the advantage of being able to identify which experimental design method yields good accuracy with the smallest number of experiments and to identify the statistical properties of an appropriate response surface estimated through data analysis. The functional relationship between the input parameters $x_{1},\;x_{2},x_{3},\;\cdots x_{k}$ and the output parameter y is expressed by Equation (3) and it can be expressed as a second order regression model as shown in Equation (4) if the predicted value of a welding factor, i.e., an output parameter, has a linear relationship with an input parameter when considering the prediction capability of linear and nonlinear models.
(3)$$Y_{i}=f{\left(x_{1},x_{2},x_{3}\right)}_{\;}$$
(4)$$Y_{i}=\beta_{0}+\overset{k}{\underset{i=1}{\sum }}\beta_{i}k_{i}+\overset{k}{\underset{i\leq j}{\sum }}\beta_{ij}x_{i}x_{j}+\epsilon$$

Equation (4) can be rewritten as Equation (5) by the least squares method.
(5)$$\hat{Y_{i}}=\hat{\beta_{i}}+\overset{k}{\underset{i=1}{\sum }}\hat{\beta_{i}}k_{i}+\overset{k}{\underset{i\leq j}{\sum }}\hat{\beta_{ij}}x_{i}x_{j}+c$$

In this study, Equation (5) can be expanded as Equation (6) since the number of input parameters is three, that is,$\;k=3_{\;}$.
(6)$$\hat{Y_{i}}=\hat{\beta_{0}}+\hat{\beta_{1}}x_{1}+\hat{\beta_{2}}x_{2}+\hat{\beta_{3}}x_{3}+\hat{\beta_{11}}x_{1}^{2}+\hat{\beta_{22}}x_{2}^{2}+\hat{\beta_{33}}x_{3}^{2}\;+\hat{\beta_{12}}x_{1}x_{2}+\hat{\beta_{13}}x_{1}x_{3}+\hat{\beta_{23}}x_{2}x_{3}$$

Here, $\hat{Y_{i}}$ is the estimated amount of welding properties, $x_{i}$ is the code unit of input parameters (welding process parameters and mechanical properties), $\hat{\beta_{0}},\;\hat{\beta_{i}},\;\hat{\beta_{ij}}$ are the least squares estimates of $\beta_{0},\;\beta_{i},\;\beta_{ij}$, respectively, and *ϵ* is the error. To develop the above regression model Equation (6), related data must be obtained through many experiments.

To obtain related data through experiments like this, trial and error in numerous experiments and economic loss may occur. To reduce these losses, a full factorial design was adopted among the response surface analysis methods of an experimental design method that well reflects the second order regression model. Most experiments for welding quality improvement involve several variables. Full factorial designs are used in such situations. Specifically, by a factorial experiment we mean that in each complete trial or replicate of the experiment all possible combinations of the levels of the factors are investigated. The coefficient of each term was calculated using Minitab. This program is software developed by Pennsylvania State University in the U.S. in 1972 and is widely used in various fields through data analysis such as engineering, sociology, business administration, and quality control. The mathematical prediction models for penetration shape (penetration width, penetration depth), upper hardness, heat-affected zone hardness (HAZ Hardness), and solidification crack susceptibility (*P_SC_*) developed using the regression coefficient and Equation (6) can be expressed as Equation (7) to Equation (11).
(7)$$P_{W}=8.871-2.537L-1.354D-2.463S+0.5375L^{2}+1.440D^{2}\;-0.06251LD-0.7389LS+2.556DS$$
(8)$$P_{D}=5.651+1.089L-1.154D-2.174S+0.1233L^{2}-0.5567D^{2}\;+0.2800LD-1.356LS+0.7222DS$$
(9)$$H_{U}=-660.3+761.7S-190.7P_{W}+314.2P_{D}-7.524P_{W}^{2}-23.84P_{D}^{2}\;+78.21SP_{W}-158.6SP_{D}+29.12P_{W}P_{D}$$
(10)$$H_{H}=97.64+202.1S-38.62P_{W}+83.79P_{D}-2.850P_{W}^{2}-6.157P_{D}^{2}\;+19.67SP_{W}-42.32SP_{D}+7.325P_{W}P_{D}$$
(11)$$P_{SC}=-249611-2196P_{D}+32.07H_{U}+1343H_{H}-4.870P_{D}^{2}-0.0227H_{U}^{2}\;-1.812H_{H}^{2}+0.4013P_{D}H_{U}+5.723P_{D}H_{H}-0.06144H_{U}H_{H}$$

To check the prediction ability of a developed mathematical prediction model, the average value of measured welding factors for each experimental condition and the predicted welding factors are compared and the error range is shown in Figure 10. The predicted model error, as shown in Table 11, mostly showed reliable results.

In addition, the variance analysis result of the predicted model confirmed a high coefficient of determination up to 96.3% at a penetration depth and a minimum 71.1% coefficient of determination at the upper hardness of the welding part. This means that the total variation of welding factors can be predicted by the coefficient of determination and the interaction, as well as the independent influence of input parameters that affect the regression model, are considered simultaneously. In the case of the developed prediction model, the coefficients are calculated using the average value of the measured data, meaning the result for the median value of the variation in the data. The uncertainty of the predicted values calculated in Equations (7)–(11) is difficult to provide because the accuracy of the mathematical model decreases when the number of measurements of the input variable used in the equation is different and errors for multiple variables are included.

### 5.2. Optimization for Welding Process of 9% Ni Steel

The MOO (Multi-Objective Optimization) algorithm applied in this study is a technique to search for nondominant solutions by imitating the evolutionary process of an organism in an optimization problem with multiple purposes. The technique has the advantage of being able to derive an optimal solution effectively by comparing and evaluating the derived nondominant solutions and explaining the trade-off relationship between objective functions. To analyze a multipurpose optimization problem, the weighted sum method was applied. First, a Pareto optimal set $P_{0}$, which is a set of nondominant solutions $x_{i}$, is created in a target space based on the mathematical definition of Pareto domination as shown in Equation (12). Genes belonging to the Pareto optimal set $P_{0}$, i.e., decision vectors, are randomly generated as many as the number of populations in the decision space, and the decision vector is evaluated by the target vector to calculate the fitness. Then, a group with a high level of nondomination and the best fitness is generated to calculate a crowding distance. It is assumed that an optimal solution set with a larger crowding distance secures the diversity of solutions, and then a multipurpose optimal solution is derived [23,24,25].
(12)$$\forall i\in \{1,\;2,\;3,\;\cdots ,\;n\}:f_{i}\left(a\right)\leq f_{i}\left(b\right)\wedge \exists j\in \left\{1,\;2,\;3,\;\cdots ,\;n\right\}:f_{i}\left(a\right)\leq f_{i}{\left(b\right)}_{\;}{}_{\;}$$

Based on the above theorem, a flow chart of the MOO optimization method is shown in Figure 11, and MATLAB, a commercial numerical analysis program, was used to apply and modify the optimization method. To optimize the welding process parameters where solidification crack susceptibility occurred, 162 data points in Table 9, that were learned in discriminant analysis, were used again and the parameters and levels to drive the MOO optimization technique are shown in Table 12. The fitness factor to implement the best driving performance of the MOO algorithm is applied in various ways by increasing the population size in the program from 50 to 100 in increments of 10. The selection of this population size accurately predicted the welding process and examined whether it was possible to calculate variables suitable for regulatory conditions. In the MOO technique, the range of fiber laser welding process parameters was selected as min. [3 kW, −0.5 mm, 0.5 m/min] to max. [5 kW, +0.5 mm, 0.8 m/min] and a multipurpose optimization problem that considers solidification crack susceptibility as an index to evaluate the quality deterioration characteristics of a welding part that occur in 9% Ni steel in the process parameter range was analyzed.

The objective function is a mathematical model of the characteristics of a system and the constraints represent the conditions that the system parameters can have. Therefore, Equations (13) and (15) show an objective function $f{\left(x\right)}_{\;}$ of an arbitrary system having $x_{\;}$ as a parameter as well as the constraints and ranges necessary to optimize this function [26]. In Equation (14), $g{\left(x\right)}_{\;}$ is the constraints function that can derive the optimization interval within the input range of the solidification crack susceptibility.
(13)$$\mathrm{Optimize}\;f{\left(L,\;D,\;S\right)}_{\;}$$
(14)$$g{\left(L,\;D,\;S\right)}_{\;}$$
(15)$$P_{\mathrm{SC}}<132.2_{\;}$$

Based on the optimization procedure as defined above, tests 2, 4, 14, and 17 were selected to follow the MOO algorithm. The respective welding process parameters, expected welding factors, and discriminant results, which were modified to satisfy the constraints by the optimization procedure, are shown in Table 13.

Figure 12 shows the solidification crack susceptibility obtained by applying the modified input parameters, and the effectiveness of 9% Ni steel welding process optimization was confirmed by performing a comparative analysis with the solidification crack susceptibility generated by the existing input parameters. As a result, it was found that all four raw data points selected in the fiber laser welding process satisfied the value of 132.2 or less, which is the limiting condition for solidification crack susceptibility. In addition, the quality deterioration characteristics observed with the existing process parameters are resolved by the modified process parameters.

## 6. Conclusions

This purpose of this study was to optimize a welding process for 9% Ni steel, which is predominantly used in the LNG storage tank industry. The occurrence of solidification crack susceptibility was defined for the process, which is learned in a discriminant function, and the process parameters that cause solidification crack susceptibility were optimized. As a result, the following conclusions were obtained.
(1)The appropriate weldability of a welding part was confirmed by measuring the penetration shape, mechanical strength, and chemical composition of the welding part derived through the fiber laser welding test. The solidification crack susceptibility phenomenon was found where the upper hardness is lowered by the impurities concentrated on the upper part of the welding part. In addition, because it was difficult to secure a stable upper hardness when an index of solidification crack susceptibility of 132.2 or more was calculated, this number was defined as a criterion at which quality deterioration occurs.(2)To determine the solidification crack susceptibility of 9% Ni steel caused by welding process parameters, the SVM technique was used to learn the quality deterioration characteristics. It was then examined whether the group where quality deterioration occurred was accurately identified. As a result, it was found that the group where the solidification crack susceptibility occurred was predicted accurately at 100% and this result was used as a procedure to determine the quality deterioration of a welding part.(3)A mathematical predicted model for the response surface method was developed to apply an objective function to optimize the welding process parameters where quality deterioration occurs and it was used in a multipurpose optimization algorithm. By entering raw data from where the solidification crack susceptibility occurred into the optimization algorithm created by the defined objective function and constraints, the quality deterioration characteristics intrinsic in the process parameters were supplemented.(4)The predicted welding factors were calculated by entering the input parameters supplemented with the quality deterioration characteristic into the response surface mathematical model. When re-entering the output parameters into a discriminant system, it was found that the possibility of quality deterioration of all raw data, where the solidification crack susceptibility is considered, was removed.

## Figures and Tables

**Figure 1 materials-13-05617-f001:**
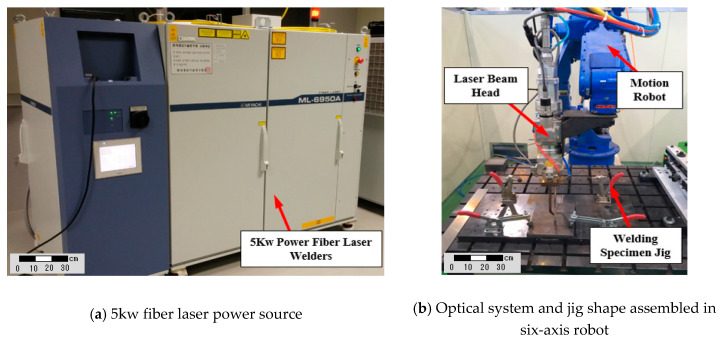
Experimental setup for fiber laser welding.

**Figure 2 materials-13-05617-f002:**
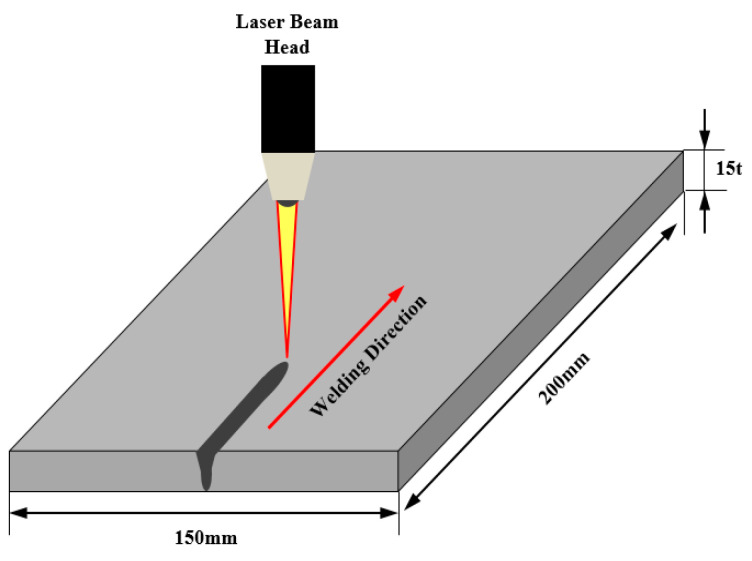
A schematic diagram for the fiber laser welding process.

**Figure 3 materials-13-05617-f003:**
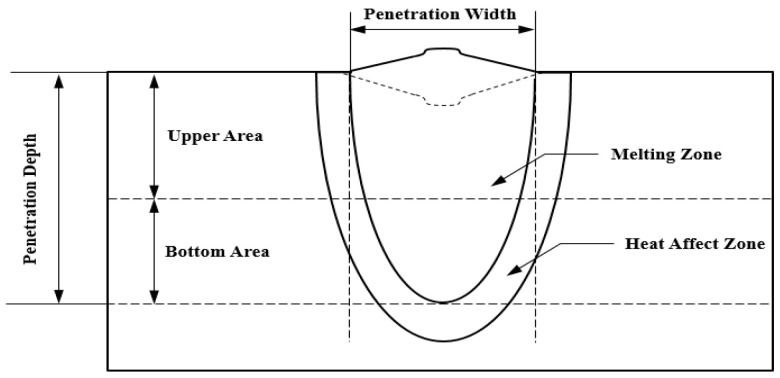
A schematic diagram of penetration geometry.

**Figure 4 materials-13-05617-f004:**
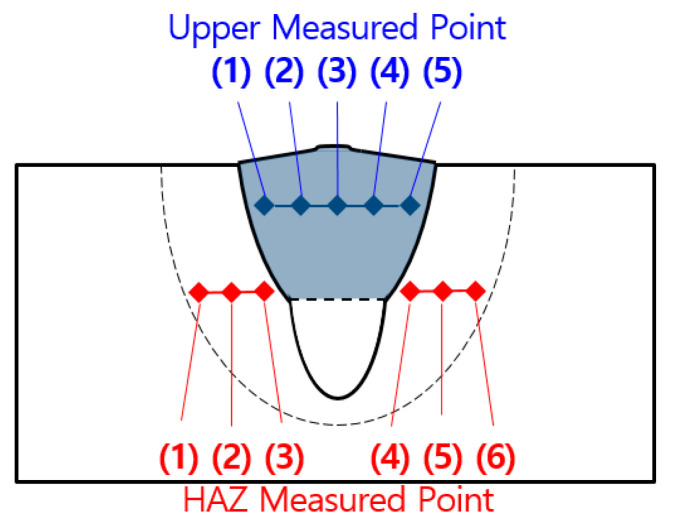
A schematic diagram of the hardness test.

**Figure 5 materials-13-05617-f005:**
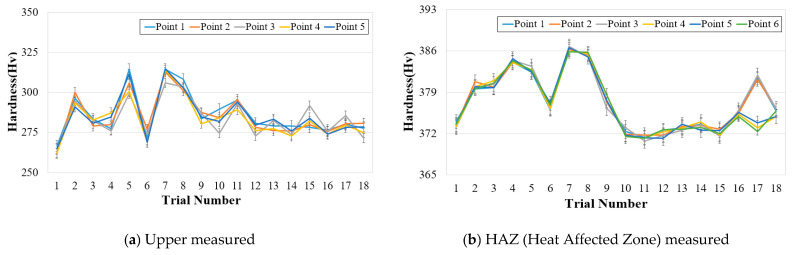
Hardness distributions according to measured position.

**Figure 6 materials-13-05617-f006:**
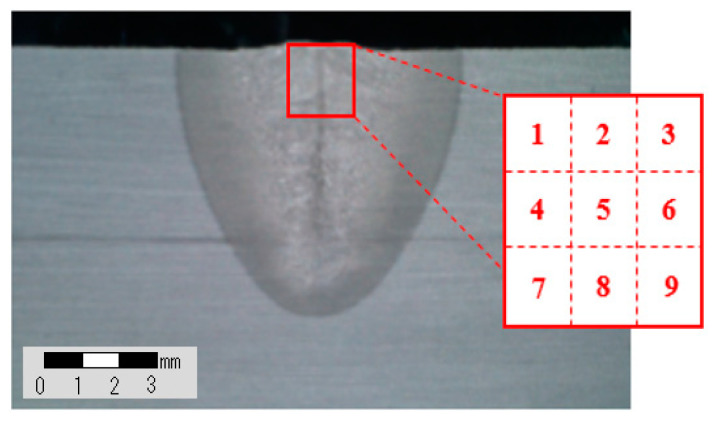
Definition of measurement section for weldment.

**Figure 7 materials-13-05617-f007:**
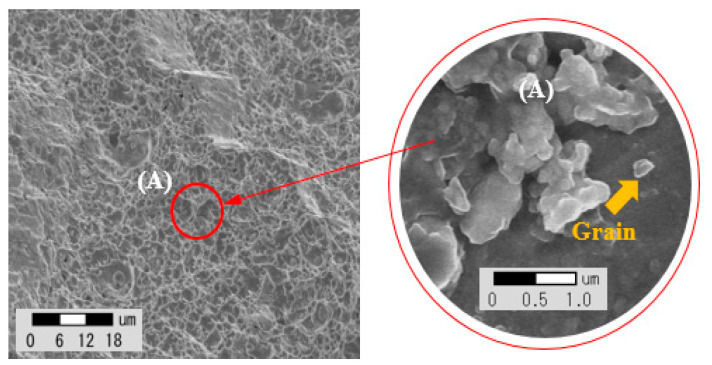
Result of SEM image for upper area in weldment.

**Figure 8 materials-13-05617-f008:**
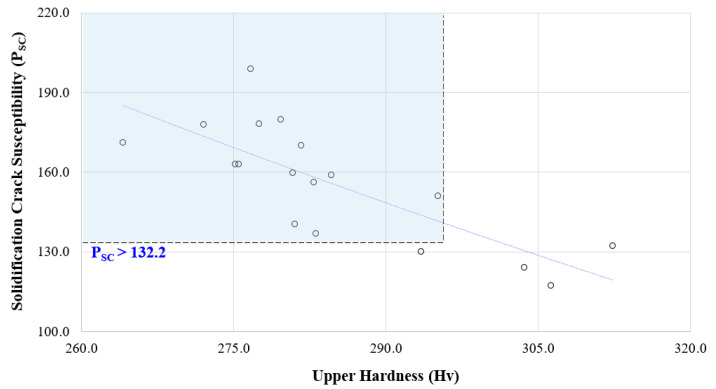
*P_SC_* distributions according to upper hardness in fiber laser welding.

**Figure 9 materials-13-05617-f009:**
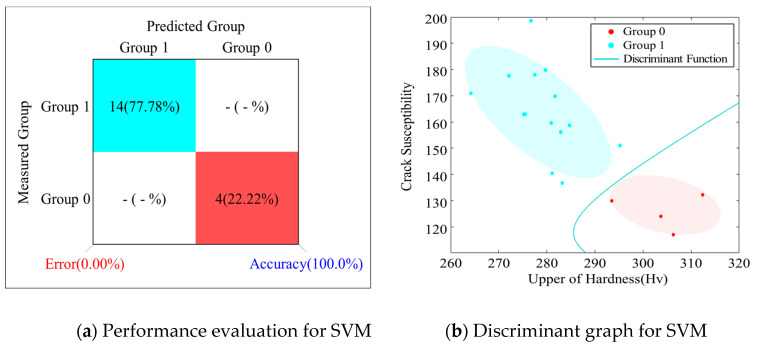
Solidification crack susceptibility discriminant in fiber laser welding: (**a**) Performance evaluation for SVM and (**b**) Discriminant graph for SVM.

**Figure 10 materials-13-05617-f010:**
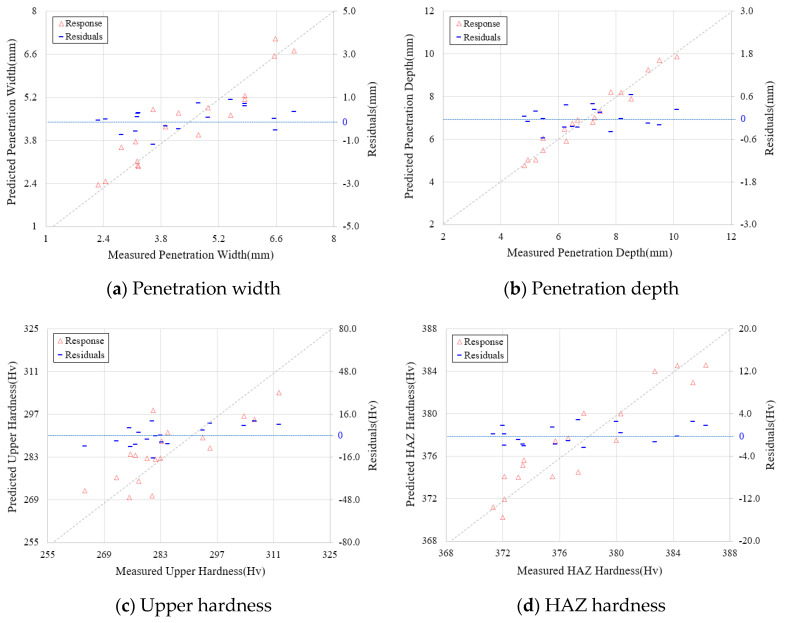
Comparison between measured and predicted welding factors according to mathematical model: (**a**) Penetration width, (**b**) Penetration depth, (**c**) Upper hardness, (**d**) HAZ hardness and (**e**) Crack susceptibility.

**Figure 11 materials-13-05617-f011:**
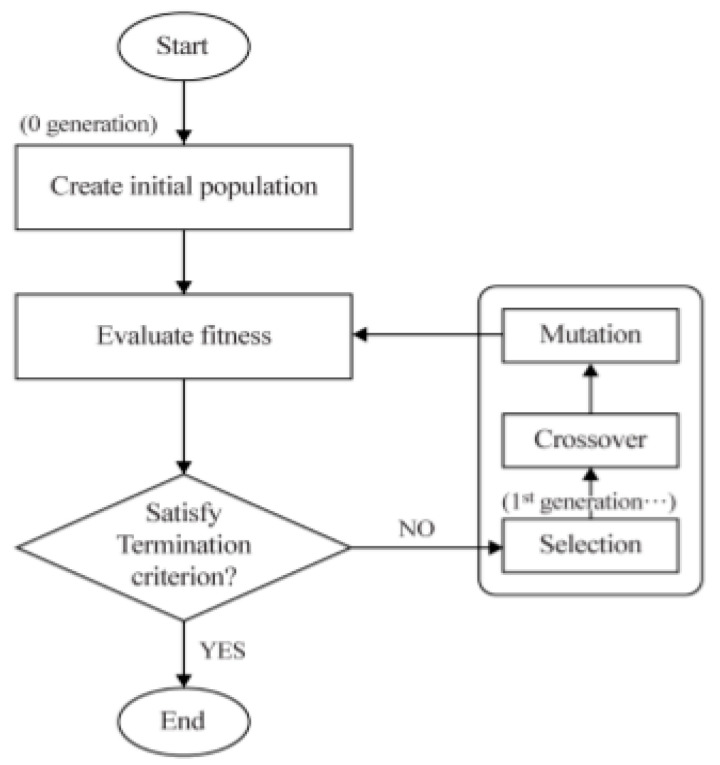
A flow chart for the Multi-Objective Optimization (MOO) method to predict welding parameters.

**Figure 12 materials-13-05617-f012:**
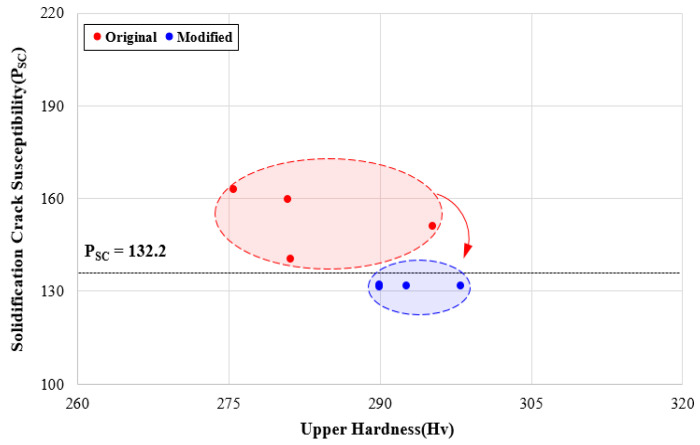
Crack susceptibility distributions using modified input parameters.

**Table 1 materials-13-05617-t001:** Mechanical properties of base metal.

Material	Yield Strength (MPa)	Tensile Strength (MPa)	Elongation (%)	Hardness (HV)
A553-1	651.6	701.1	26.6	243

**Table 2 materials-13-05617-t002:** Chemical composition of base metal.

Component	C	Si	Mn	S	P	Ni	Fe
Percentage (wt.%)	0.05	0.67	0.004	0.003	0.25	9.02	Bal.

**Table 3 materials-13-05617-t003:** Fiber laser welding parameters and their levels.

**Parameter**	**Symbol**	**−1**	**0**	**1**
Laser Power (kW)	L	3.0	4.0	5.0
Defocusing (mm)	D	−0.5	0.0	0.5
Welding Speed (m/min)	S	0.5	−	0.8
Fixed Parameter	Wavelength: 1070 nm			
	Optical Fiber Diameter: 200 µm			
	Shielding Gas Flow Rate: 18 L/min, (L/min)			

**Table 4 materials-13-05617-t004:** Experimental plan of fiber laser welding process.

Test No.	*L*	*D*	*S*	Test No.	*L*	*D*	*S*
1	3.0	−0.5	0.5	10	3.0	−0.5	0.8
2	3.0	0.0	0.5	11	3.0	0.0	0.8
3	3.0	0.5	0.5	12	3.0	0.5	0.8
4	4.0	−0.5	0.5	13	4.0	−0.5	0.8
5	4.0	0.0	0.5	14	4.0	0.0	0.8
6	4.0	0.5	0.5	15	4.0	0.5	0.8
7	5.0	−0.5	0.5	16	5.0	−0.5	0.8
8	5.0	0.0	0.5	17	5.0	0.0	0.8
9	5.0	0.5	0.5	18	5.0	0.5	0.8

**Table 5 materials-13-05617-t005:** Results of fiber laser welding experiment.

Test No.	Penetration Width (mm)				Penetration Depth (mm)				Penetration Geometry
	1st	2nd	3rd	Average	1st	2nd	3rd	Average	
1	3.93	3.90	3.90	3.91 ± 0.008	6.49	6.47	6.51	6.49 ± 0.009	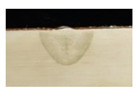
2	3.19	3.18	3.17	3.18 ± 0.005	6.64	6.66	6.64	6.65 ± 0.005	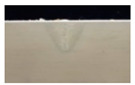
3	4.73	4.72	4.69	4.71 ± 0.010	7.21	7.22	7.15	7.19 ± 0.018	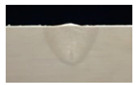
4	5.82	5.86	5.84	5.84 ± 0.009	8.52	8.51	8.55	8.53 ± 0.010	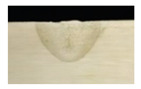
5	5.48	5.49	5.49	5.49 ± 0.003	8.17	8.15	8.15	8.16 ± 0.005	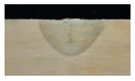
6	3.61	3.71	3.5	3.61 ± 0.050	7.84	7.82	7.79	7.82 ± 0.012	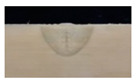
7	6.59	6.58	6.58	6.58 ± 0.003	9.11	9.12	9.11	9.11 ± 0.003	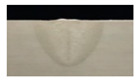
8	6.54	6.55	6.55	6.55 ± 0.003	9.49	9.51	9.53	9.51 ± 0.009	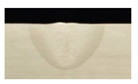
9	7.01	7.03	7.04	7.03 ± 0.007	10.09	10.09	10.11	10.1 ± 0.005	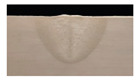
10	2.51	2.47	2.37	2.45 ± 0.034	4.86	4.78	4.79	4.81 ± 0.021	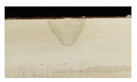
11	2.21	2.28	2.32	2.27 ± 0.026	4.95	4.89	4.95	4.93 ± 0.016	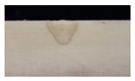
12	3.26	3.27	3.22	3.25 ± 0.012	5.19	5.23	5.21	5.21 ± 0.009	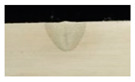
13	3.25	3.23	3.17	3.22 ± 0.020	5.49	5.48	5.44	5.47 ± 0.012	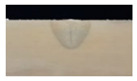
14	3.22	3.30	3.20	3.24 ± 0.025	6.25	6.24	6.29	6.26 ± 0.012	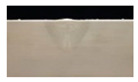
15	2.82	2.84	2.86	2.84 ± 0.009	5.43	5.44	5.54	5.47 ± 0.029	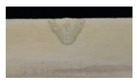
16	4.94	4.97	4.91	4.94 ± 0.014	6.18	6.24	6.21	6.21 ± 0.014	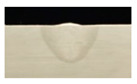
17	4.25	4.19	4.21	4.22 ± 0.014	7.26	7.24	7.24	7.25 ± 0.005	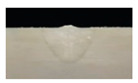
18	5.84	5.83	5.85	5.84 ± 0.005	7.47	7.41	7.44	7.44 ± 0.014	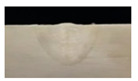

**Table 6 materials-13-05617-t006:** Results of hardness according to welding process and parameters.

Test No.	Upper by Point (Hv)						Test No.	HAZ (Heat Affected Zone) by Point (Hv)						
	1st	2nd	3rd	4th	5th	Avg.		1st	2nd	3rd	4th	5th	6th	Avg.
1	266.6	262.1	264.9	262.3	264.8	264.2	1	374.1	373.4	373.1	373.2	373.6	373.7	373.5
2	296.4	300.0	294.6	293.6	291.1	295.2	2	379.6	380.8	379.8	380.0	380.0	379.6	380.0
3	283.3	279.0	282.6	282.7	280.9	281.7	3	379.8	379.9	380.4	381.0	379.8	380.6	380.3
4	277.4	279.8	276.1	287.3	284.7	281.1	4	384.1	384.1	384.4	384.0	384.7	384.3	384.3
5	314.6	305.9	299.2	300.6	311.4	306.3	5	382.4	382.8	383.4	382.8	382.4	382.7	382.7
6	270.7	277.1	274.2	269.5	269.0	272.1	6	376.1	376.5	376.5	376.3	377.2	376.8	376.6
7	314.6	312.7	306.0	313.7	314.7	312.4	7	386.3	386.3	386.8	386.0	386.6	385.9	386.3
8	308.3	301.1	303.3	302.8	302.8	303.7	8	385.4	385.4	385.0	385.6	385.0	385.8	385.4
9	283.7	287.5	287.1	280.6	284.6	284.7	9	377.5	377.4	376.3	378.5	377.6	378.5	377.7
10	289.6	284.2	274.6	284.3	281.8	282.9	10	372.4	371.9	373.0	371.5	371.8	371.5	372.0
11	295.3	295.6	293.0	289.3	294.2	293.5	11	371.4	371.8	370.8	371.3	371.3	371.2	371.3
12	280.9	278.1	272.8	276.3	279.5	277.5	12	372.6	371.8	371.7	372.3	371.2	372.7	372.1
13	279.2	276.2	282.4	277.4	283.2	279.7	13	373.0	373.2	372.6	373.0	373.6	372.8	373.1
14	279.1	276.1	273.7	272.8	276.1	275.5	14	373.6	373.4	373.5	374.0	372.6	373.1	373.4
15	278.2	279.8	291.8	282.1	284.0	283.2	15	371.9	372.8	371.6	371.8	372.5	372.0	372.1
16	276.5	275.7	275.7	274.1	274.0	275.2	16	375.7	375.5	376.0	375.3	375.6	374.9	375.5
17	280.3	280.4	285.7	279.6	278.3	280.9	17	381.4	381.2	381.9	373.0	373.8	372.4	377.3
18	277.6	280.9	271.5	274.9	278.6	276.7	18	375.9	376.3	376.2	374.9	374.9	375.8	375.7

**Table 7 materials-13-05617-t007:** Results of chemical properties for fiber laser welding.

Test No.	Ti	Nb	Mo	Si
1	0.0204 ± 8.0 × 10^−5^	0.1036 ± 8.3 × 10^−4^	0.0154 ± 9.1 × 10^−5^	0.7395 ± 7.0 × 10^−2^
2	0.0205 ± 1.3 × 10^−4^	0.1038 ± 1.0 × 10^−3^	0.0157 ± 1.3 × 10^−4^	0.6729 ± 6.5 × 10^−2^
3	0.0203 ± 8.5 × 10^−5^	0.1040 ± 1.1 × 10^−3^	0.0156 ± 1.2 × 10^−4^	0.7357 ± 7.3 × 10^−2^
4	0.0205 ± 9.5 × 10^−5^	0.1049 ± 8.6 × 10^−4^	0.0159 ± 1.6 × 10^−4^	0.6374 ± 7.1 × 10^−2^
5	0.0205 ± 1.0 × 10^−4^	0.1031 ± 8.4 × 10^−4^	0.0160 ± 1.2 × 10^−4^	0.5598 ± 5.2 × 10^−2^
6	0.0205 ± 9.7× 10^−5^	0.1048 ± 7.9 × 10^−4^	0.0157 ± 1.1 × 10^−4^	0.7617 ± 6.8 × 10^−2^
7	0.0206 ± 8.9 × 10^−5^	0.1051 ± 9.7 × 10^−4^	0.0157 ± 1.7 × 10^−4^	0.6101 ± 5.7 × 10^−2^
8	0.0206 ± 8.4 × 10^−5^	0.1055 ± 9.6 × 10^−4^	0.0158 ± 1.5 × 10^−4^	0.5828 ± 6.4 × 10^−2^
9	0.0204 ± 1.2 × 10^−4^	0.1030 ± 7.2 × 10^−4^	0.0156 ± 9.3 × 10^−5^	0.6989 ± 7.9 × 10^−2^
10	0.0205 ± 6.7 × 10^−5^	0.1067 ± 6.7 × 10^−4^	0.0159 ± 1.4 × 10^−4^	0.6899 ± 5.9 × 10^−2^
11	0.0207 ± 6.3 × 10^−5^	0.1059 ± 7.2 × 10^−4^	0.0157 ± 1.3 × 10^−4^	0.6023 ± 6.7 × 10^−2^
12	0.0205 ± 1.1 × 10^−4^	0.1032 ± 8.0 × 10^−4^	0.0162 ± 1.2 × 10^−4^	0.7631 ± 6.8 × 10^−2^
13	0.0204 ± 1.0 × 10^−4^	0.1043 ± 7.4 × 10^−4^	0.0157 ± 1.3 × 10^−4^	0.7689 ± 6.4 × 10^−2^
14	0.0206 ± 6.3 × 10^−5^	0.1052 ± 1.1 × 10^−3^	0.0158 ± 1.4 × 10^−4^	0.7125 ± 5.7 × 10^−2^
15	0.0205 ± 7.6 × 10^−5^	0.1043 ± 7.9 × 10^−4^	0.0156 ± 1.6 × 10^−4^	0.6254 ± 5.9 × 10^−2^
16	0.0205 ± 8.9 × 10^−5^	0.1043 ± 1.2 × 10^−3^	0.0158 ± 1.3 × 10^−4^	0.7126 ± 5.7 × 10^−2^
17	0.0204 ± 7.5 × 10^−5^	0.1041 ± 1.0 × 10^−3^	0.0153 ± 1.1 × 10^−4^	0.7018 ± 7.0 × 10^−2^
18	0.0206 ± 8.8 × 10^−5^	0.1054 ± 6.2 × 10^−4^	0.0159 ± 1.3 × 10^−4^	0.8315 ± 5.0 × 10^−2^

**Table 8 materials-13-05617-t008:** Solidification crack susceptibility data for discriminant analysis in fiber laser welding.

Test No.	Upper Hardness (Hv)	Value of *P_SC_*	Crack Susceptibility	Test No.	Upper Hardness (Hv)	Value of *P_SC_*	Crack Susceptibility
1	264.2 ± 0.8	170.9 ± 2.1	Unstable	10	282.9 ± 2.2	156.1 ± 1.8	Unstable
2	295.2 ± 1.3	151.0 ± 2.0	Unstable	11	293.5 ± 1.0	129.9 ± 2.0	Stable
3	281.7 ± 0.7	169.8 ± 2.2	Unstable	12	277.5 ± 1.3	178.0 ± 2.0	Unstable
4	281.1 ± 1.9	140.4 ± 2.1	Unstable	13	279.7 ± 1.2	179.8 ± 1.9	Unstable
5	306.3 ± 2.7	117.0 ± 1.6	Stable	14	275.5 ± 1.0	162.9 ± 1.7	Unstable
6	272.1 ± 1.4	177.6 ± 2.1	Unstable	15	283.2 ± 2.1	136.7 ± 1.8	Unstable
7	312.4 ± 1.5	132.2 ± 1.7	Stable	16	275.2 ± 0.4	162.9 ± 1.7	Unstable
8	303.7 ± 1.1	124.0 ± 1.9	Stable	17	280.9 ± 1.1	159.6 ± 2.1	Unstable
9	284.7 ± 1.1	158.7 ± 2.4	Unstable	18	276.7 ± 1.4	198.6 ± 1.5	Unstable

**Table 9 materials-13-05617-t009:** Learning data for discriminant of fiber laser welding quality.

Test No.	*L*	*D*	*S*	*P* _W_	*P* _D_	*H* _U_	*H* _H_	*P* _SC_	Group
1	3.0	−5.0	0.5	3.91 ± 0.008	6.49 ± 0.009	264.2 ± 0.8	373.5 ± 0.1	170.9 ± 2.1	Unstable
2	3.0	0.0	0.5	3.18 ± 0.005	6.65 ± 0.005	295.2 ± 1.3	380.0 ± 0.2	151.0 ± 2.0	Unstable
3	3.0	5.0	0.5	4.71 ± 0.010	7.19 ± 0.018	281.7 ± 0.7	380.3 ± 0.2	169.8 ± 2.2	Unstable
4	4.0	−5.0	0.5	5.84 ± 0.009	8.53 ± 0.010	281.1 ± 1.9	384.3 ± 0.1	140.4 ± 2.1	Unstable
5	4.0	0.0	0.5	5.49 ± 0.003	8.16 ± 0.005	306.3 ± 2.7	382.7 ± 0.1	117.0 ± 1.6	Stable
6	4.0	5.0	0.5	3.61 ± 0.050	7.82 ± 0.012	272.1 ± 1.4	376.6 ± 0.1	177.6 ± 2.1	Unstable
7	5.0	−5.0	0.5	6.58 ± 0.003	9.11 ± 0.003	312.4 ± 1.5	386.3 ± 0.1	132.2 ± 1.7	Stable
8	5.0	0.0	0.5	6.55 ± 0.003	9.51 ± 0.009	303.7 ± 1.1	385.4 ± 0.1	124.0 ± 1.9	Stable
9	5.0	5.0	0.5	7.03 ± 0.007	10.1 ± 0.005	284.7 ± 1.1	377.7 ± 0.3	158.7 ± 2.4	Unstable
10	3.0	−5.0	0.8	2.45 ± 0.034	4.81 ± 0.021	282.9 ± 2.2	372.0 ± 0.2	156.1 ± 1.8	Unstable
11	3.0	0.0	0.8	2.27 ± 0.026	4.93 ± 0.016	293.5 ± 1.0	371.3 ± 0.1	129.9 ± 2.0	Stable
12	3.0	5.0	0.8	3.25 ± 0.012	5.21 ± 0.009	277.5 ± 1.3	372.1 ± 0.2	178.0 ± 2.0	Unstable
13	4.0	−5.0	0.8	3.22 ± 0.020	5.47 ± 0.012	279.7 ± 1.2	373.1 ± 0.1	179.8 ± 1.9	Unstable
14	4.0	0.0	0.8	3.24 ± 0.025	6.26 ± 0.012	275.5 ± 1.0	373.4 ± 0.2	162.9 ± 1.7	Unstable
15	4.0	5.0	0.8	2.84 ± 0.009	5.47 ± 0.029	283.2 ± 2.1	372.1 ± 0.2	136.7 ± 1.8	Unstable
16	5.0	−5.0	0.8	4.94 ± 0.014	6.21 ± 0.014	275.2 ± 0.4	375.5 ± 0.1	162.9 ± 1.7	Unstable
17	5.0	0.0	0.8	4.22 ± 0.014	7.25 ± 0.005	280.9 ± 1.1	377.3 ± 1.7	159.6 ± 2.1	Unstable
18	5.0	5.0	0.8	5.84 ± 0.005	7.44 ± 0.014	276.7 ± 1.4	375.7 ± 0.2	198.6 ± 1.5	Unstable

*L:* Laser Power (kW);·*D*: Defocusing (mm);·*S*: Welding Speed (m/min); *P_W_*: Penetration Width (mm); *P*_D_: Penetration Depth (mm); *H*_U_: Upper Hardness (Hv); *H*_H_: HAZ Hardness (Hv);·*P*_SC_: Solidification Crack; Susceptibility.

**Table 10 materials-13-05617-t010:** Results of group discriminant for crack susceptibility according to SVM.

Test No.	Measured Group	Predicted Group	Test No.	Measured Group	Predicted Group
1	1	1 (1.00)	10	1	1 (1.00)
2	1	1 (1.00)	11	0	0 (0.27)
3	1	1 (1.00)	12	1	1 (1.00)
4	1	1 (0.99)	13	1	1 (1.00)
5	0	0 (0.00)	14	1	1 (1.00)
6	1	1 (1.00)	15	1	1 (1.00)
7	0	0 (0.00)	16	1	1 (1.00)
8	0	0 (0.00)	17	1	1 (1.00)
9	1	1 (1.00)	18	1	1 (1.00)

**Table 11 materials-13-05617-t011:** Analysis variance tests for predicted model for welding factors.

Design Parameter	Predicted Model	SE (Standard Error)	R^2^ (Coefficient of Determination, %)
*P* _W_	Response Surface Analysis	0.769	86.4
*P* _D_	Response Surface Analysis	0.423	96.3
*H* _U_	Response Surface Analysis	10.83	71.1
*H* _H_	Response Surface Analysis	2.541	80.7
*P* _SC_	Response Surface Analysis	2.414	87.3

**Table 12 materials-13-05617-t012:** MOO algorithm parameters and their values.

Optimal Method		MOO (Multi-Objective Optimization)
Range of Local Parameters	*L* (Laser Power)	[−0.5 ≤ Input ≤ +0.5] kW
	*D* (Defocusing)	[−0.25 ≤ Input ≤ +0.25] mm
	*S* (Welding Speed)	[−0.15 ≤ Input ≤ +0.15] m/min
Range of Constraints	*P*_SC_ (Crack Susceptibility)	*P*_SC_ ≤ 132.2
Fitness Factor	Population Size	50, 60, 70, 80, 90, 100
Solver		Constrained Nonlinear Minimization
Algorithm		Trust Region Reflective Algorithm
Derivatives		Gradient Supplied

**Table 13 materials-13-05617-t013:** Results of welding parameters modified by optimization process.

Test No.	Original			Modified			Welding Factors					Group
	*L*	*D*	*S*	*L*	*D*	*S*	*P_W_*	*P_D_*	*H_U_*	*H_H_*	*P_SC_*	
2	3.0	0.0	0.5	3.45	−0.24	0.49	4.2	7.5	289.9	381.9	132.1	Stable
4	4.0	−5.0	0.5	3.91	−0.51	0.51	5.0	7.7	289.9	382.4	131.5	Stable
14	4.0	0.0	0.8	3.84	−0.08	0.86	2.5	5.3	298.0	374.5	131.8	Stable
17	5.0	0.0	0.8	5.23	0.24	0.92	4.7	6.4	292.6	379.4	131.7	Stable

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
