# Peer review of "A Study on the Algorithm of Quality Evaluation for Fiber Laser Welding Process of ASTM A553-1 (9% Nickel Steel) Using Determination of Solidification Crack Susceptibility"

_materials, 2020, doi:10.3390/ma13245617_

Round 1
Reviewer 1 Report
The ppaer submitted to MATERIALS treats about the laser welding of the ASTM-A553 steel.
The paper is written in good english and easy to read.
After reading that article my first thought is that Materials is not a good journal for it. The paper is typical "manufacturing" paper not material. There not many materials issues arised not the investigations made exept hardness and microstructures. The typical "materials science chain" Material-microstructure-property is not shown. Authors wrote "nickel based alloy is composed of austenite structure.... " this steel is not austenitic, if authors had something else in mind those sentences are making a lot of confusion. Other thing is that this is not a welding but remelting of the plate- and this is huge difference, I would expect the weld quality examination based on the title. Also what is crystalisation coeficient-?
I definietly suggest to reject here and transfer to some manufacturing journal.
Author Response
Thank you for your high opinion.
Please check attached file.

Reviewer 2 Report
Cool work. The impact of LNG in metals will be a big issue in the coming years. Some figures need to big more clear. Maybe more modern characterization methods need to be used in the future for mechanical characterization, for example: A machine learning approach to fracture mechanics problems, X Liu, CE Athanasiou, NP Padture, BW Sheldon, H Gao, Acta Materialia 190, 105-112
The authors could cite more related work. Other than that, I suggest the manuscript for publication.
Author Response

(The authors gave the same response as above.)

Reviewer 3 Report
It seems the authors carried out a lot of experiments. I feel the paper could still be improved. I suggest a major revision. My comments are listed as follows,
- I feel the title is not appropriate. A lot of things were discussed in this paper rather than the crack susceptibility.
- There were a few grammar mistakes. Please double check.
- The authors presented the melting zone geometry data and the hardness data. However, the effect of the processing parameters on these characteristics were not discussed in detail. How did the processing parameters influence the melting depth, width, and hardness?
- What was the purpose of investigation on the HAZ hardness.
- I noticed that the A553-1 steel itself doesn’t contain the elements such as Ti, Nb,Mo. Moreover, there was no filling materials during the welding process. However, the authors detected these elements in the welding zone and presented these data on Page 7. Where did these elements come from? Also, I notice the fraction of these values were very low. I am thinking if the EDS is good enough to detect the elements with such a small quantity.
- The authors divided the processing parameters into stable and unstable group. What did the “stable” and “unstable” here mean?
- The R-square of hardness of upper region was only 71.1, which was not a very strong fitting.
- How did the criterion of 132.2 was selected.
- Had the authors run experiments by using the modified processing parameters?
Author Response

(The authors gave the same response as above.)

Reviewer 4 Report
- Line 14: "order" should be "orders".
- Line 14: You use the abbreviation "LNG" without defining it. I know it is a relatively common term, but please define it when you first use it. Additionally, I suggest you avoid abbreviations in the abstract.
- Line 17: "is limiting" should be "limits".
- Line 22: "identification of tendency" should be "identification of the tendency".
- Lines 20-24: this sentence is very long and difficult to understand. I recommend that the authors divide it into at least two sentences.
- Line 32: You use the abbreviation LNG without defining it. I accept that you define it on line 33, but please define all abbreviations when they are first used.
- Line 35: there is a minor issue with the spacing before "LNG storage tanks". Please fix this.
- Line 36: In "QT(Quenching", there should be a space before the parenthesis. This error is also made in other places where a parenthesis is used, e.g. line 41.
- Line 49-53: The sentences describing Kim's work and Lee's work are not particularly useful. Rather than saying what they did, please tell the readers what these studies concluded - as a reader, I am not interested that Kim "diagnosed the problems of prior studies", I want to know what these problems were, what studies had these problems and how Kim overcame these problems. The sentences in lines 53-59 are much better.
- Lines 60-80: It is frustrating for the reader that many concepts are explored here, but no citations are given to sources of more detailed information. Please add appropriate citations.
- Lines 64-66: "The research results are being used for strong welding, high speed, and automation, etc. in industries by complementing the shortcomings of the latest welding technology." gives very little information. What aspects of the technologies are being complemented by each other? Please give specific examples to help the reader understand the need for your current study.
- Lines 71-72: "the concentration of solute elements in a residual melt occurs" does not make sense. You can either say : "the concentration of solute elements in a residual melt increases" or "a concentration of solute elements in a residual melt occurs".
- Line 83: There is a missing space between "5kW" and "class".
- Figure 1: Please add an approximate scale bar to each image. Please also label the subfigures as (a) and (b) and describe each part more exactly in the caption.
- Tables 1 and 2: please provide the uncertainty for each measurement, with a description of the significant sources of uncertainty.
- Line 100: There is a missing space between "[11]." and "Figure 3".
- Lines 104-110: This paragraph is unnecessary complicated and contains some terms that are not widely understood by the scientific/engineering community, such as "Full Factorial Design". I suggest that you remove such terminology and simply describe what you really mean. For example, that you selected three values of laser output, three levels of defocusing and two welding speeds. You then tried every possible combination of these parameters and that you did this to try to detect any interactions between the parameters.
- Using complicated language makes your paper more difficult to read and some readers will trust the results less since they might feel you are trying to make your article look more impressive than it really is. I suggest you do not let them think that and let your study impress them!
- Table 3: The symbols "L", "D", and "S" are technically mathematical variables and so they must be formatted as such, using an equation environment in Word/LaTeX.
- Table 5: Please provide a sketch that shows how the penetration width and penetration depth are defined. Also, there should be a space between the words in your column headings and your units "(mm)".
- Table 6: It is very difficult to understand the meaning of Table 6. I suggest you plot a graph instead, as this will be much easier for readers to understand and see the important effects. Also, please give uncertainties (error bars on a graph) for each measurement point, with a description of how these uncertainties were calculated.
- Line 142: Again, there is a missing space between "EDS" and "(Energy Dispersive Spectroscopy)".
- Figure 5: How large was each of the nine zones?
- Figure 6a: You must present a scale bar for the micrographs. EDX (EDS) results come form a region that is at least several micrometres in each direction, due to spreading of the electron beam inside the sample. Therefore, you results can only be associated with the impurities if they are several micrometers in size. If not, the results represent a mixture of the impurity and the sample matrix.
- Figure 6b: the spectrum is not useful for the readers - it gives no information. I suggest you remove it. Furthermore, the table is only relevant for one sample, which is not identified anywhere. You should identify this sample and include the table as a separate table in the document, not a subfigure (i.e. the table currently part of Figure 6b should become Table 7 and the current Table 7 should become Table 8, etc.). Again, you must provide uncertainties of all measurements - this is usually available in the EDX software itself.
- Table 7: Please provide uncertainties, as described in the previous point.
- Lines 159-162: This should be presented the background, not the discussion. The discussion should compare your finding to those available in literature.
- Lines 171-172: please describe more precisely what you mean by "the hardness of the upper welding part was stabilised". What does stabilised mean in this context? How did you decide that this occurred below 130 and not below 140, which could also be argued, based on the appearance of Figure 7.
- Figure 7: missing spaces before parentheses, as describe in previous comments. Please add error bars to the graph. I accept that you do not have all the information to add error bars in the y-direction, but you can use the uncertainties in your measured compositions to provide a lower bound of the uncertainty in the "Solidification Crack Susceptibility" values. You should also explain in the text if Equation (1) has no uncertainties given in the source.
- What is the line that is drawn on this plot? Is is a best fit line? If so, please give the equations of this line and some assessment of the goodness-of-fit (please note that R2 alone is not sufficient as a measurement of goodness of fit).
- Line 178: How did you decide on the limit of 132.2? Is this decided elsewhere? If so, you must provide a citation. If not, you must explain how that value was selected.
- Line 193: this is also advanced statistics terminology that may not be understood by many of the readers of this paper. I suggest you describe the SVM technique in simpler language.
- Equation (2): you must define each of the variables, w, x and b in the text, before Equation 2 is presented.
- Line 197: Please define the abbreviation "HAZ" (I know it is obvious to most metallurgists, but your paper might be read by people without a background in metallurgy and they should be told what "HAZ" stands for, so they can find out more information if they need to.
- Line 198: the "P" in "PSC" should be italic - it is a mathematical variable.
- Table 9: You have formatted the variables "L", "S" and "D" correctly here. However, the subscripts in "Hu", "HH" and "PSC" should not be italic - they are labels, not variables.
- Figure 8: the plot is unreadable - the text is too small and the image is pixelated. Please replete the data with larger text, so it can be read and understood.
- Line 228: a space is needed between "is," and "k = 3".
- Line 232: ε does not appear anywhere in your equations, yet you have defined it in the text. Please check your equations and/or remove ε form the text.
- Line 235: please explain what a "complete factor design" is.
- Line 237: please explain what "MINITAB" is. Is is a computer program?
- Equations 7-12: please provide uncertainties for each derived value. The fact that you measured your values to four significant figures makes it extremely unlikely that six significant figures is appropriate now, when these measurements were used to derive the fitted values. However, this must be assessed for each value individually using statistical methods.
- Lines 252-253: what is "crystallization coefficient"?
- Table 11: R2 on its own is not suitable as a goodness-of-fit measurement. It is only a measurement of correlation. I accept that it is often used to measure goodness-of-fit, but it is wrong. R2 can be used to measure goodness of fit to some extent, but only if taken together with a consideration of the residuals of the fit. Other statistical tools are preferable.
- Figure 9: please plot these data again with larger text - the current plots are difficult to read. Please also put error bars for each prediction and measurement.
- Line 267: there is yet another missing space before "P0".
- Table 12: Please see my previous comments for formatting of mathematical variables.
- Equation 13: You cannot write the word "Optimize" in this way - it implies that it is a product of different mathematical variables ("O", "p", "t", etc.. Please make this normal text (i.e. not italic).
- Line 285: missing space after "0.8m/min]".
- Equation 14: the variable g was never defined i the text. Please define how g is different to f.
- How was the fitness factor defined fro the MOO algorithm?
- Did you try any welds with the parameters derived by the MOO algorithm? If not, it would be very good if you could do so and add it to this paper. If not, I suggest you list it as recommended future work.
- Line 323: missing space after "100%".
- The article claims to optimise "the fiber laser welding process by determining the solidification crack susceptibility of ASTM A553-1". To meet this goal, you must provide some optimum set of parameters. You do not do this. Please either provide an optimised set of parameters or remove references to optimisation form the paper.
Author Response

(The authors gave the same response as above.)

Round 2
Reviewer 1 Report
Authors implemented changes of reviewers. I think now it may be published.
Author Response
Thank you for your opinions. They were very helpful for us.
Reviewer 3 Report
My questions and comments were well addressed. I suggest accept the paper of the current version.
Author Response
Thank you for your high comments. They were very helpful for us.
Reviewer 4 Report
Thank you for the changes you have made. I believe the manuscript will be suitable for publication and it is now improved. However, I do have som remaining points about which I would like to be satisfied before I can recommend that the manuscript is accepted for publication.
- There are many instances where you use a parenthesis "(" without a space before it. I am not aware of any language where that is allowed and it must be corrected before the paper is accepted for publication. I raised tins as a problem in my previous comments and you have done it again when you added new parentheses and even kept some instances of it from the original submission. Please fix this.
- I still believe that figure 1 would be enhanced by adding a scale bar. It is obvious that it is a photograph and not a sketch or micrograph, but it is not possible to tell how big either of the machines is. Is the machine in figure 1a one metre wide or five metres? Is figure 1b showing some region 20 cm wide or 1 m wide?
- If you are not able to provide any uncertainty (even a reasonable estimate would be useful if the exact uncertainties are not available) for the values, please say so in the caption of each table and include the reason why no uncertainty is available.
- In table 3, the "D" is still not formatted correctly, although the other variables are.
- In table 4, the variables are not formatted correctly.
- The subscripts in the variables in Table 9 are still incorrect. Pleas read my previous comment about it carefully and make the change to the subscripts. They should not be italic. Both Word and LaTeX can do this for you inside an equation environment and instructions for this are widely available on the internet.
- The new figure 5 is much better, thank you! However I suggest you make the text bigger, as it is difficult to read when printed. As a guide, the printed text in the figure should be the same size as the main text in the manuscript.
- In figure 6 (previously figure 5), the size of the zones cannot be understood by the reader, as the micrograph lacks a scale bar. Please add one. Also, the scale in figure 7 is extremely difficult to read. I suggest you add a new one that is clearer.
- I accept your reasoning about my comment 26.
- I am still not clear about your response to my point number 27. You cannot pick a threshold value simply to be convenient - it must either have a physical basis or be based on the data (i.e. come form regression or a similar analytical technique). It is not clear that your selection of 132.2 is based on anything. I guess that is is the highest score at which the samples were stable. If so, you must explain this to justify your argument. As it is written now, your argument is very difficult to follow. However, I am not happy with your use of this value to decide who the upper hardness becomes constant.
- Do the authors mean that the uncertainty in the data are less than the random scatter in the nine values they use to form an average? Is fo, why not give the error bar as the standard deviation of these nine values? If the line is a regression line, an uncertainty in the slope is easy to calculate using any analysis software (including Microsoft Excel). Do the authors have any metrics for the goodness of fit of the line? This will help support whether the input and output variables are correlated or not. It is not only needed for prediction.
- In response to my comment 29, I understand this now. Please add such an explanation not the manuscript.
- In réponse to my comment 30. This may be the case, but could you then give a simple explanation about what the algorithm does and why it is appropriate to use it in this study. There is no need to use any numbers in the explanation, just describe the purpose of the algorithm.
- Figure 9 is improved, but I suggest you make the text even larger - it is only just readable now and there is no reason it has to be so small.
- Please describe what a "full factorial design" is for the benefit of any readers not familiar with DoE.
- When I asked you to describe MINITAB, I meant that you should explain it in the text (or cite some source of information). I apologise if that was not clear.
- For equations 1-5, please explain in the manuscript why uncertainties cannot be given. If they can be given, please give them.
- I accept that R2 is often used in the way you describe, but it is (statistically) weak evidence. All it can show on its own is that two variables are correlated. It cannot be used to support any particular equation of a trendline. If you have the output data, you can calculate a more suitable statistic fairly easily using any spreadsheet or analysis program. Alternatively, I suggest that you also consider the residuals: for example, are they randomly positive and negative or all positive in one part of the graph?
- In response to my comment 49, the authors misunderstood the question. I know what a fitness factor means (the manuscript actually does not include the term at all). I was asking what expression was used to calculate the fitness factor - this usually has a very strong influence on the final results and it is often simply the choice of the researchers who perform the modelling.
- In response to my comment 50, I believe figure 11 is the output of a calculation. I asked if you performed any experiments to check the effects of the parameters derived by MOO. Please correct me if I have misunderstood and figure 11 contains measured data and not calculated data.
- I do not understand the response to my comment 52. My point is that if you do not present some "optimum" parameters that are generated by your analysis, you cannot claim to be studying the "optimization" of anything. If you present some new parameters, this will be OK. Alternatively, remove the words "optimization of the" from your title. That would then fit your manuscript.
Author Response
Thank you for your high opinions. They were very helpful for us.
Please check attached file.
